# Sleep Quality and Associated Factors in Adults with Type 2 Diabetes: A Retrospective Cohort Study

**DOI:** 10.3390/ijerph18063025

**Published:** 2021-03-15

**Authors:** Ching-Pyng Kuo, Shu-Hua Lu, Chien-Ning Huang, Wen-Chun Liao, Meng-Chih Lee

**Affiliations:** 1School of Nursing, Chung Shan Medical University, Taichung 40201, Taiwan; pyng@csmu.edu.tw; 2Department of Nursing, Chung Shan Medical University Hospital, Taichung 40201, Taiwan; 3School of Nursing, China Medical University, Taichung 406040, Taiwan; shuhua@mail.cmu.edu.tw; 4Department of Nursing, China Medical University Hospital, Taichung 404332, Taiwan; 5Department of Internal Medicine, Chung Shan Medical University Hospital, Taichung 40201, Taiwan; cshy049@csh.org.tw; 6Institute of Medicine, Chung Shan Medical University, Taichung 40201, Taiwan; 7Department of Nursing, Asia University, Taichung 41354, Taiwan; 8Department of Family Medicine, Taichung Hospital, Ministry of Health and Welfare, Taichung 40343, Taiwan; 9Institute of Population Health Sciences, National Health Research Institutes, Miaoli 35053, Taiwan; 10College of Management, Chaoyang University of Technology, Taichung 413310, Taiwan

**Keywords:** sleep, diabetes mellitus, type 2, diabetes complications

## Abstract

*Purpose:* Sleep disturbance is one of the major complaints among patients with diabetes. The status of diabetes control and associated complications may contribute to sleep disturbance. This study explored night time sleep and excessive daytime sleepiness in adults with type 2 diabetes and examined the association of diabetes control and associated complications on their sleep quality. *Methods:* A retrospective cohort study design was used. Type 2 diabetic patients (87 females and 79 males, aged 63.1 ± 10.5 years) were recruited from the outpatient clinics of the endocrine department. Sleep quality was assessed by the Pittsburg Sleep Quality Index and the Epworth Sleepiness Scale. Diabetes control and complications were obtained by retrospectively reviewing patients’ medical records over 1 year prior to study enrollment. *Results:* 72.3% of recruited patients had poor glycemic control, and 71.1% had at least one diabetic complication. 56.0% of patients experienced poor sleep quality, and 24.1% had excessive daytime sleepiness. Those who were female (OR = 3.45) and who had ophthalmological problems (OR = 3.17) were associated with poor night time sleep quality, but if they did exercise to the point of sweating (OR = 0.48) reduced the risk of poor sleep quality. Furthermore, poor sleep quality (OR = 4.35) and having nephropathy (OR = 3.78) were associated with a higher risk of excessive daytime sleepiness. *Conclusions:* Sex, ophthalmological problems, nephropathy, and no exercise to the point of sweating are associated with sleep problems in patients with type 2 diabetes. Both lifestyle behaviors and diabetic complications affect sleep disturbances in patients with diabetes.

## 1. Introduction

Diabetes mellitus (DM), one of the most common chronic medical diseases encountered in clinical practice, is a major public health challenge. Its prevalence continues to increase, and it affects approximately 422 million individuals worldwide [1]. A growing body of evidence indicates that impaired sleep is associated with the development of DM [2,3], and interestingly, sleep disturbances are commonly observed in this patient population. Studies have reported that >50% of patients with type 2 diabetes (T2DM) showed sleep disturbances [4,5,6]. Of these 50%, 8% had prolonged sleep latency, 23% had poor sleep maintenance, 26% had both prolonged sleep latency and poor sleep maintenance [4], and 11.3% had excessive daytime sleepiness [6,7]. Poor sleep quality and excessive daytime sleepiness are commonly observed in patients with T2DM. Impaired sleep is known to disturb optimal glycemic control and is associated with diabetic complications [8,9]. The association between sleep and DM is bidirectional [10], and both important issues to consider. Further studies are warranted to investigate contributors to sleep disorders in patients with T2DM to achieve better glycemic control in these patients.

Many studies have compared patients with T2DM and healthy controls to evaluate sleep quality in the former. It was observed that elevated serum glycated hemoglobin (HbA1C) levels were associated with lower sleep efficiency and more moving time [8], and altered glucose metabolism may adversely affect sleep quality [11]. However, few studies report the factors associated with sleep in patients diagnosed with T2DM. A correlational study performed by Lamond et al. (2000) investigated three physical correlates of DM as predictors of sleep disruption in patients with T2DM and reported that nocturia and neuropathic pain were associated with sleep onset and maintenance [4]. Other DM-related factors such as polyuria, retinopathy, and psychological issues may also affect sleep quality in patients with DM [12]. These findings are reported by cross-sectional studies and suggest that glycemic control may affect sleep duration and quality in patients diagnosed with DM. In addition to DM, lifestyle behaviors and sleep environment are known to affect sleep quality [13]. In the current study, we retrospectively investigated the association between glycemic control and diabetic complications and sleep-related factors to gain a deeper understanding of the potential predictors of sleep disorders in patients with DM. We evaluated nighttime sleep quality and excessive daytime sleepiness in patients with T2DM and investigated the potential effect of glycemic control and associated complications monitored over the year prior to study enrollment, as well as lifestyle behaviors and sleep environment on patients’ current sleep quality.

## 2. Method

### 2.1. Study Design and Participants

We used a convenient sampling technique for this retrospective cohort study. Patients were recruited from the outpatient clinic of the endocrinology department of a medical center in central Taiwan. Patients diagnosed with T2DM, not hospitalized, and aged 18–85 years were eligible for inclusion in this study. Those who had (1) major mental disorders such as psychosis or severe depression or anxiety (Hospital Anxiety and Depression Scale [HADS] score > 15) [14]; (2) cognitive impairment (Mini-Mental State Examination [MMSE] score ≤ 25 in individuals with middle to high education levels or ≤19 in illiterate individuals or those with low education levels) [15]; or (3) a history of alcohol or drug abuse were excluded. We contacted 276 patients; however, 5 patients refused to participate, and 105 were excluded from the analysis because of incomplete laboratory data. Eventually, 166 patients completed the study. No differences were observed between the included and the excluded patients regarding sex, education levels, and duration of DM. However, the excluded patients were younger (60.0 ± 10.1 years).

### 2.2. Ethical Considerations

This study was approved by the Institutional Review Board of the CSMUH (No: CS09025) and was performed in accordance with the Declaration of Helsinki of 1995. All participants provided written informed consent for the research, and all patient data were anonymized.

### 2.3. Data Collection

We obtained data regarding demographics, self-reported sleep patterns, and lifestyle behaviors including smoking, alcohol consumption, exercise, and sleep environment (noise and disruption). The status of diabetes control (blood glucose, lipid levels, and blood pressure [BP]) and complications (neuropathy, nephropathy, ophthalmological complications, and vascular disease) [16] were assessed by reviewing patients’ medical records. Data were observed and measured retrospectively over 1 year from the study enrollment.

### 2.4. Measures

#### 2.4.1. Demographic Data and Duration of Diabetes

A self-administered questionnaire was used to obtain data regarding patient characteristics. Age, sex, education levels, marital status, occupation, and interval since diagnosis of DM (in years) were recorded. Body mass index was calculated by body weight (kg) divided by body height squared (m^2^), and classified based on the recommendation of the WHO [17] and the Taiwan Ministry of Health and Welfare [18].

#### 2.4.2. Sleep Quality

The Pittsburg Sleep Quality Index (PSQI) [19] was used to assess habitual nighttime sleep quality over 1 month. The PSQI consists of 19 self-rated questions that are used to generate specific scores representing the following 7 components of the PSQI: Subjective sleep quality, sleep latency, sleep duration, habitual sleep efficiency, sleep disturbances, use of sleep medication, and sleepiness-induced daytime dysfunction. The component scores were summed to obtain a global PSQI score ranging between 0 and 21; higher scores indicated poorer sleep quality. A score of 5 was used as the cut-off value to categorize sleep quality as “good” (≤5) or “poor” (>5). Cronbach’s alpha in this study was 0.78.

Excessive daytime sleepiness was assessed using the Epworth Sleepiness Scale Chinese version (ESS) [20,21]. Sleepiness assessed across 8 daytime situations was rated from 0 = no dozing to 3 = high risk of dozing, with a total score ranging between 0 and 24. A higher score indicated excessive daytime sleepiness. A cut-off score of 10 was used to indicate a risk of sleep apnea (ESS ≥ 10) [21] in this study. The Cronbach’s alpha in this study was 0.78.

#### 2.4.3. Control of Diabetes

Targets for diabetes control suggested by the American Diabetes Association Guidelines 2018 [22] are as follows: (1) Optimal glycemic control: HbA_1_C (normal levels <7%, abnormal levels ≥7%); (2) BP control: Systolic BP (normal levels <130 mmHg, abnormal levels ≥130 mmHg) and diastolic BP (normal levels <80 mmHg, abnormal levels ≥80 mmHg); (3) lipid control: Low-density lipoprotein-cholesterol (normal levels <100 mg/dL), high-density lipoprotein-cholesterol (normal levels >60 mg/dL, abnormal levels <40 mg/dL [males], <50 mg/dL [females]), and serum triglycerides (normal levels <150 mg/dL, abnormal levels >150 mg/dL). These data were obtained after retrospectively reviewing patients’ medical records over 1 year prior to study enrollment.

#### 2.4.4. Complications of Diabetes

Diabetic complications, including those associated with vascular disease (hypertension, coronary artery disease, and stroke), ophthalmological issues (retinopathy, cataract, and glaucoma), neuropathy (trembling and diabetic feet), and nephropathy (urine albumin-to-creatinine ratio ≥ 30 mg albumin/g creatinine, microalbuminuria ≥ 30 mg albumin/g creatinine, and macroalbuminuria ≥ 300 mg albumin/g creatinine) [16] were retrospectively retrieved from the medical records of each patient.

#### 2.4.5. Lifestyle Behaviors and Sleep Environment

Patients’ current status of smoking, alcohol consumption, and exercise were evaluated. Patients were interviewed to obtain information regarding each parameter as follows: (1) Exercise (Do you exercise? How often and how long do you exercise? When you exercise, do you exercise to the point of sweating?); (2) Smoking habits (Do you smoke currently?); and (3) Alcohol consumption (Do you drink? How often do you drink per week?). Exercise to the point of sweating was defined as performing any exercise to a level that induces sweating and was used for data analysis. Smoking was defined as yes or no currently. Alcohol consumption was defined as never, sometimes (<once a week), and often (>every other day) in this study.

Sleep environment was assessed using two items on the visual analogue scale, including the noise level and disrupted perception during sleep (ranging from 0 [very quiet or no disruption] to 10 [extremely noisy or extremely disrupted]). Noisy and disturbed sleep environment was defined as the score of the analogue scale greater than 5.

#### 2.4.6. Anxiety and Depression

Anxiety and depression were assessed using the Hospital Anxiety and Depression Scale (HADS) [14]. The HADS contains 7 items each for anxiety and depression subscales with a score of 0 (not at all) to 3 (most of the time) for each question. The two subscales were summed to generate overall anxiety and overall depression scores, with a total score of 21. A subscale score of ≥11 was interpreted as potential anxiety or depression. The Cronbach’s alpha values for anxiety and depression scales were 0.86 and 0.85, respectively.

### 2.5. Data Analysis

Descriptive statistics, including frequency, percentage, mean, and standard deviation (SD) were used to understand the sample distribution of sleep quality and diabetes control. Normality of the dependent variables was tested by using the Kolmogorov–Smirnov test. Inferential statistics, including the odds ratio (OR), the Pearson correlation coefficient, and multiple linear or logistic regression analysis were used to determine the factors associated with sleep. A two-tailed *p* value < 0.05 was considered statistically significant. The SPSS for Windows software, version 26.0 (SPSS Inc., Chicago, IL, USA) was used for all statistical analyses.

## 3. Results

### 3.1. Basic Patient Characteristics

Patient demographics are presented in Table 1. This study included 166 patients (79 men and 87 women) with T2DM, aged 33–86 years (63 ± 10.5 years), who voluntarily participated in this research. The mean duration of DM since diagnosis was 10.1 ± 7.9 years (range 0.5–51 years). Most patients did not currently smoke (87.3%) or consume alcohol (83.8%) and reported some form of exercise (51.8%). All patients were at a low risk of anxiety and depression.

### 3.2. Status of Diabetes Control, Complications, Nighttime Sleep, and Excessive Daytime Sleepiness

The status of diabetes control and complications in patients with T2DM are presented in Table 2. Assessment of diabetes control included evaluation of glucose, BP, and lipid levels; 72.3% of patients showed poor glycemic control (HbA_1_C ≥ 7%), 70.5% showed elevated BP, 87.3% showed dyslipidemia, and 98.8% showed at least one abnormality in glucose, BP, or lipid levels. This group of patients was classified as those with poor diabetes control. Regarding diabetic complications, >50% of the patients (56.6%) had vascular disease, followed by ophthalmological complications and neuropathy, and 71.1% had at least one complication (Table 2).

Table 3 shows the distribution of sleep status and excessive daytime sleepiness in patients with DM. Most patients reported <7 h of nighttime sleep (67.5%), 29.5% reported nighttime sleep latency >30 min, and 40.4% reported <85% sleep efficiency. Naps were not included in nighttime sleep hours. We observed poor sleep quality (PSQI ≥ 5) in >50% of the patients (56.0%). In contrast, only 32.5% of patients reported >7 h of nighttime sleep and 24.1% reported excessive daytime sleepiness (ESS score ≥ 10). The major causes of disturbed nighttime sleep perceived by patients were as follows: “have to get up to use bathroom” and “can’t get to sleep within 30 min”, followed by “wake up in the middle of the night or early morning” and “cough or snore loudly”. Compared with excessive daytime sleepiness, poor nighttime sleep was the predominant complaint in patients with T2DM.

### 3.3. Factors Related to Nighttime Sleep and Excessive Daytime Sleepiness in Patients with Type 2 Diabetes

Since the normality of the PSQI and ESS score was not assumed by the Kolmogorov–Smirnov test (*p* = 0.000 for both), the univariate logistic regression with odds ratio (OR) was used to determine the association between poor nighttime sleep and excessive daytime sleepiness with patient characteristics, mental status, lifestyle behaviors, sleep environment, diabetes control, and complications (Table 4). The OR for poor nighttime sleep quality was lower in male sex and in those who exercise to the point of sweating; and was higher in those who had vascular disease, and ophthalmological complications. The greater the number of diabetic complications, the worse the nighttime sleep was, particularly in those with ophthalmological complications. Moreover, the OR of excessive daytime sleepiness was three- to four-fold in those who had poor sleep quality and nephropathy complications compared to those who did not (Table 4).

Multiple logistic regression analysis was used to determine the relative effects of patient characteristics, mental status, lifestyle behaviors, sleep environment, diabetes control, and complications on nighttime sleep quality and excessive daytime sleepiness based on the significant univariate logistic regression results. Sex, no exercise, and ophthalmological complications were significant risk factors associated with poor nighttime sleep (Table 5). The odds ratio (OR) for poor nighttime sleep risk was 81.3% lower in men than in women, 52.1% lower in those who exercised to the point of sweating than in those who did not, and 3.17-fold higher in those with ophthalmological complications than in those without such issues. Sleep quality and nephropathy were significant risk factors for excessive daytime sleepiness. The OR for excessive daytime sleepiness risk was 4.35-fold higher in those with poor sleep quality and 3.78-fold higher in those with nephropathy than in those without this complication.

## 4. Discussion

### 4.1. Contributors to Poor Nighttime Sleep Quality

We observed that >50% of the patients included in this study (56%) reported poor nighttime sleep based on a PSQI cut-off score of 5. This result concurs with that of previous studies, which report poor sleep quality in >50% of patients with DM [5,6,23]. Waking up to use the bathroom during the night (≥three times per week) was the leading cause of sleep disturbances reported by 50% of the patients in this study. These findings are similar to previous studies of high prevalence (66.4%) of sleep disorders with nocturia (35.4% to 80.4%) in type 2 diabetes [24,25,26]. Frequent episodes of nocturia lead to frequent awakening with consequently impaired sleep quality [24]. Sleep fragmentation secondary to nocturia and frequent nighttime urination may reflect poor glycemic control [27] and diabetic complications of albuminuria, higher serum creatinine, high-sensitivity C-reactive protein levels [28], and microvascular complications [25]. “Unable to fall asleep within 30 min” and “waking up in the middle of the night or early morning” were the second and third leading causes of sleep disturbances in this study, reported by 30.7% and 20.5% of patients with DM, respectively, which was also consistent with a previous study [24]. Nocturia, and difficulties in sleep initiation and maintenance were the main sleep disturbances in patients with DM and poor sleep quality and may lead to poor quality of life [24] and increased mortality [26]. Target related factors to manage poor sleep quality in diabetic patients is needed.

The results of univariate logistic regression analysis showed that poor nighttime sleep quality was associated with female sex, no exercise to the point of sweating, vascular disease, and ophthalmological complications. Multiple logistic regression analysis showed that after adjusting for age and sex, no exercise to the point of sweating and ophthalmological complications were significant risk factors (Table 5). Patients who did not engage in intensive exercise and were diagnosed with ophthalmological complications including retinopathy, cataracts, and glaucoma experienced poorer sleep quality, with an explained variance of 21.3%. This finding in patients with ophthalmological complications is consistent with that reported by previous studies, which describe the role of pathogenesis of retinopathy in sleep deprivation [29]. Positive association of retinopathy, but not nephropathy or neuropathy, with nocturia in Japanese patients with type 2 diabetes mellitus [25] may also indicate the important role of retinopathy on nocturia and poor sleep in patients with diabetes. With regard to optimum sleep, light is the predominant environmental signal (zeitgeber) that synchronizes the body’s circadian rhythm [30]. The photoreceptive capacity of the intrinsically photoreceptive retinal ganglion cells (ipRGCs) in the eyes is dependent on the expression of melanopsin to mediate the effect of light on circadian wake-sleep functions [30]. Therefore, retinopathy in patients with DM affects the function of the ipRGCs and consequently affects sleep.

With regard to the effect of exercise on sleep, previous studies have reported positive associations between self-reported exercise levels and sleep duration [31]. Exercise mediates the effects of sleep on energy balance-related health outcomes [32]. Intensive physical activity levels were also independently associated with a lower incidence of diabetic retinopathy in Japanese patients with DM [33]. Muscle contractions of the lower extremities during exercise promote venous return to the heart [34], which helps to alleviate symptoms of nocturia and improves sleep quality. Clinicians should assess patients’ physical activity levels and perform thorough evaluation for ophthalmological complications before prescribing sleep medications in patients with DM presenting with poor sleep quality. Since tolerance in exercise intensity for each patient is different, a tailored exercise program is necessary to concern. Exercise to the point of sweating regardless of the intensity and preventing ophthalmological complications are important in improving sleep quality in patients with DM.

### 4.2. Contributors to Excessive Daytime Sleepiness

In this study, 24.1% of patients reported excessive daytime sleepiness (ESS ≥ 10) and 18.1% reported sleep disturbed by snoring or coughing ≥thrice a week. Excessive daytime sleepiness is less common than poor nighttime sleep in patients with T2DM. However, in this study, we observed that the prevalence of excessive daytime sleepiness (based on ESS scores) was higher than the prevalence of 11.3–21.7% reported in the literature [6,35,36]. Notably, these rates were lower than 40.9% (the rate observed with the Berlin Questionnaire assessment) [37]. Excessive daytime sleepiness is a risk factor for sleep apnea and narcolepsy [38], and its prevalence is higher in patients with DM than in those without DM [37]; therefore, assessment of daytime sleepiness is important in these patients. It affects patients’ energy and motivation levels, which consequently affect their ability to effectively manage their chronic condition.

The results of univariate logistic regression analysis in this study showed that poor nighttime sleep quality and nephropathy were associated with excessive daytime sleepiness. After adjusting for age and sex, poor sleep quality and nephropathy were significant risk factors for excessive daytime sleepiness in patients with DM. Excessive daytime sleepiness is often attributed to a sedentary lifestyle or laziness; however, it is noteworthy that this condition may be a manifestation of serious disorders such as DM [36]. Poor nighttime sleep quality is a primary contributor to excessive daytime sleepiness. However, excessive daytime sleepiness is also a potential risk factor for sleep apnea (obstructive or central type). Cytokines and insulin resistance may mediate excessive daytime sleepiness and sleep apnea [39,40]. Intermittent desaturations and recurrent hypoxia further deteriorated the diabetic outcomes [40]. Previous studies have reported a positive association between excessive daytime sleepiness and glycemic levels in patients with DM [36]. However, in this study, we observed a significant association between excessive daytime sleepiness and nephropathy. A previous study that examined sleep apnea syndrome in type 2 diabetes also revealed a similar association of obstructive sleep apnea syndrome and the development of diabetic nephropathy [41]. Excessive daytime sleepiness in patients with DM may represent fatigue due to long-term complications of nephropathy, which is attributable to oxygen desaturation [42] and impaired oxygen transport secondary to anemia, chronic inflammation, and nutritional deficiencies [43]. These conditions are known to result in low energy levels with exhaustion and represent symptoms of daytime excessive sleepiness. Patients with DM and concomitant nephropathy experience persistent fatigue; therefore, appropriate interventions are necessary to prevent or treat nephropathy to improve patients’ energy levels.

### 4.3. Diabetes Control and Complications

Management of patients with T2DM includes control of hyperglycemia, hypertension, and hyperlipidemia, and prevention of microvascular complications (nephropathy, retinopathy, and neuropathy) [44]. In the current study, 72.3% of patients with DM showed poor glycemic control; this rate is significantly higher than that observed in other Asian countries [45]. Studies have emphasized the importance of compliance with medications, diet, and exercise for optimal glycemic control. The association between exercise to the point of sweating and nighttime sleep quality observed in this study also highlights the importance of intensive exercise in managing DM and improving patients’ sleep. Not just performing exercise, but considering the intensity of exercise tailored to an individual level to induce beneficial effect. Exercise to the point of sweating is a good and easy indicator for assessing exercise intensity, and can be recommended for patients. However, only 50% of the patients in this study performed intensive exercise. Improving physical activity and exercise in patients with DM is an important strategy to optimize both glycemic control and sleep quality.

Age is an important factor affecting sleep quality [46]. However, age was not a significant factor on either poor night time sleep or excessive daytime sleepiness (Table 4 and Table 5) in this study, no matter continuous age, or divided into age groups with 10 years or >65 years were used in data analysis. For patients with type 2 diabetes, poor sleep quality may be dominant in all age groups. A current study of systematic review and meta-analyses on the impact of age on complications and mortality revealed that age at diabetes diagnosis was associated with risk of all-cause mortality and complications; each 1 year increase in age at diabetes diagnosis was associated with a 3% to 5% decreased risk of all-cause mortality and complications [47]. Younger age at diabetes diagnosis is associated with a higher risk of complications and mortality, and is not a protecting factor after diabetes diagnosed. Early evaluation and sustained interventions to manage sleep are essential in all ages of type 2 diabetes patients.

### 4.4. Limitations of the Study

Following are the limitations of this study: (a) The retrospective design of this longitudinal study is a drawback. Although we retrospectively retrieved data regarding patients’ status of diabetes control one year before evaluation of their sleep quality, we could not conclusively establish a causal association between glycemic control and diabetic complications and patients’ poor sleep. (b) We used only self-reported data regarding sleep hours and sleep quality; therefore, we cannot exclude a perception bias affecting sleep evaluation. Notably, objective sleep measures such as actigraphy or polysomnography can only measure the duration of sleep (hours) and its distribution and not sleep quality. Sleep quality is the “degree of excellence or satisfaction” experienced with sleep. Individual variations are observed in the amount of sleep in various situations. Therefore, sleep duration only serves as one of the indicators of sleep, and sleep quality as an individualized indicator of sleep is more reliable to assess an individual’s sleep needs. (c) This study was performed at a single diabetes center; therefore, the results of this single-center study may not be widely generalizable.

## 5. Conclusions

Poor nighttime sleep and excessive daytime sleepiness are common in patients with T2DM; in this study, >50% of patients experienced poor sleep. Female sex, exercise to the point of sweating, and ophthalmological complications were potentially associated with nighttime sleep quality. Health care providers should thoroughly screen patients with DM and concomitant poor sleep for complications, particularly ophthalmological issues. Prevention of retinopathy and other ophthalmological complications are important to manage patients’ poor sleep. Notwithstanding the onset of complications, exercise to the point of sweating should be emphasized (regardless of exercise patterns) to manage DM and improve sleep quality. Screening for nighttime sleep and excessive daytime sleepiness should be considered an integral component of holistic diabetes care for early detection and treatment. Health care providers should adopt a broader perspective and focus on expanding their knowledge regarding the role of sleep and sleep-induced factors in patients with T2DM and also create greater awareness regarding the role of good sleep quality among these patients.

## Figures and Tables

**Table 1 ijerph-18-03025-t001:** Personal characteristics, mental status, and life style behaviors (N = 166).

Variable	M ± SD	Range	*n*	%
Personal characteristics				
Age	63.1 ± 10.5	33–86		
31–49			18	10.8
50–64			69	41.6
65–74			53	31.9
≥75			26	15.7
Sex				
Female			87	52.4
Male			79	47.6
Occupation				
No			61	36.8
Retired			50	30.1
Yes			55	33.1
Education				
Illiterate			18	10.8
Elementary and middle school			88	53.0
High school and higher			60	36.2
Marriage				
Married			122	73.5
Single/widower/widow			44	26.5
DM duration	10.1 ± 7.9	0.5–51		
0.5–10 years			104	62.7
11–20 years			47	28.3
20–30 years			9	5.4
≥31 years			6	3.6
BMI	25.9 ± 3.7	17.7–36.2		
<18.5			2	1.2
18.5–24			55	33.3
24–27			48	29.1
27–30			35	21.2
≥30			25	15.2
Mental status				
Anxiety (≥11)			11	6.6
Depression (≥11)			21	12.7
Life style behaviors				
Smoking, current (yes)			21	12.7
Drinking (sometimes)			8	4.8
Drinking (often)			19	11.4
Exercise to sweating (yes)			86	51.8
Sleep Environment (0 no ~10 extremely)				
Noising (yes, >5)			18	10.8
Disrupted (yes, >5)			10	6.0

**Table 2 ijerph-18-03025-t002:** Diabetes control and complications.

Variable	*n*	%
Glycemic control		
HbA_1_C ≤ 7%	46	27.7
HbA_1_C > 7%	120	72.3
Blood pressure control		
≤130/80 mmHg	49	29.5
>130/80 mmHg	117	70.5
Lipid control ^1^		
Normal	21	12.7
Abnormal	145	87.3
Diabetes control ^2^		
0 (normal)	2	1.2
1 (one abnormal)	25	15.1
2 (two abnormal)	60	36.1
3 (three abnormal)	79	47.6
Complications		
Neuropathy ^3^	36	21.7
Vascular disease ^4^	94	56.6
Nephropathy ^5^	12	7.2
Ophthalmological problems ^6^	44	26.5
No. of complications		
0 (normal)	48	28.9
1 (one abnormal)	66	39.8
2 (two abnormal)	37	22.3
3 (three abnormal)	14	8.4
4 (four abnormal)	1	0.6

Note: ^1^ Lipid control normal: TG ≤ 150, HDL ≥ 50 (female), ≥40 (male), and LDL ≤ 100. ^2^ Diabetes control: Glycemia, hypertension, and dyslipidemia. ^3^ Neuropathy: Including trembling feet and diabetes foot. ^4^ Vascular disease: Including hypertension, coronary artery syndrome, and stroke. ^5^ Nephropathy: Urine albumin-to-creatinine ratio ≥ 30 mg albumin/g creatinine, microalbuminuria ≥ 30 mg albumin/g creatinine, and macroalbuminuria ≥ 300 mg albumin/g creatinine. ^6^ Ophthalmological problems: Including retinopathy, cataracts and glaucoma.

**Table 3 ijerph-18-03025-t003:** Night time sleep and excessive daytime sleepiness in patients with type 2 diabetes.

Variable	Mean ± SD	Median (IQR)	*n*	%
Excessive daytime sleepiness	6.3 ± 4.4	5 (3–9)		
Normal (ESS ^1^ < 10)			126	75.9
Abnormal (ESS ≥ 10)			40	24.1
Night time sleep quality	7.0 ± 4.1	6 (4–10)		
Good (PSQI ^2^ ≤ 5)			73	44.0
Poor (PSQI > 5)			93	56.0
Night time sleep hours				
>7 h			54	32.5
6–7 h			60	36.2
5–5.9 h			36	21.7
<5 h			16	9.6
Night time sleep latency				
≤15 min			66	39.8
16–30 min			51	30.7
31–60 min			25	15.0
≥60 min			24	14.5
Sleep efficiency (%)				
≥85			99	59.6
75–84			33	19.9
65–74			16	9.6
<65			18	10.9
Sleep disturbances for three or more times a week				
Have to get up to use bathroom			83	50.0
Cannot get to sleep within 30 min			51	30.7
Wake up in the middle of night or early morning			34	20.5
Cough or snore loudly			30	18.1
Night mare			22	13.3
Pain			10	6.0
Feel hot			3	1.8
Cannot breathe comfortably			2	1.2
Feel cold			2	1.2

Note: ^1^ Epworth Sleepiness Scale; ^2^ Pittsburg Sleep Quality Index.

**Table 4 ijerph-18-03025-t004:** Univariate logistic regression with odds ratio of poor night time sleep and excessive daytime sleepiness with personal characteristics, mental status, lifestyle behaviors, sleep environment, diabetes control, and complications.

	PSQ ^1^			EDS ^2^		
	OR	95% CI	*p*	OR	95% CI	*p*
Age	1.020	0.990–1.051	0.192	1.027	0.991~1.065	0.145
Sex ^3^	0.290	0.153–0.552	<0.000	0.872	0.427–1.781	0.707
BMI	1.012	0.931–1.100	0.782	1.048	0.952–1.153	0.338
DM duration (year)	1.029	0.988–1.073	0.171	1.017	0.974–1.062	0.435
PSQ ^1^			-	**4.262**	**1.822**–**9.971**	**0.001**
Anxiety ^4^	0.938	0.275–3.204	0.919	2.857	0.823–9.924	0.098
Depression ^5^	2.147	0.789–5.846	0.130	1.697	0.633–4.553	0.294
Smoking ^6^	0.845	0.338–2.115	0.719	0.486	0.136–1.746	0.269
Drinking ^7^	1.073	0.670–1.719	0.769	0.621	0.315–1.227	0.170
Exercise to sweating ^8^	**0.491**	**0.263**–**0.916**	**0.025**	0.909	0.446–1.852	0.793
Noise volume > 5	1.265	0.465–3.443	0.645	1.242	0.414–3.726	0.699
Noise disruption > 5	3.341	0.687–16.241	0.135	0.776	0.158–3.815	0.755
Glycemic control ^9^	1.098	0.555–2.175	0.789	2.113	0.860–5.192	0.103
Lipid control ^10^	2.302	0.899–5.897	0.082	1.018	0.348–2.980	0.974
BP Control ^11^	0.579	0.290–1.155	0.121	1.138	0.516–2.512	0.748
Diabetes control ^12^	0.999	0.669–1.493	0.998	1.342	0.820–2.196	0.242
Neuropathy ^13^	1.688	0.679–4.197	0.260	1.059	0.389–2.881	0.911
Vascular disease ^13^	**1.884**	**1.010–3.515**	**0.047**	1.200	0.582–2.474	0.621
Nephropathy ^13^	1.624	0.469–5.619	0.444	**3.529**	**1.070–11.647**	**0.038**
Ophthalmological problems ^13^	**3.100**	**1.437–6.690**	**0.004**	1.259	0.573–2.764	0.566
Complications ^14^	**1.740**	**1.217–2.488**	**0.002**	1.165	0.803–1.690	0.422

Note. ^1^ PSQ = Poor sleep quality, was assessed by the PSQI global score ≤ 5 = 1 = good sleepers; >5 = 2 = poor sleepers. ^2^ EDS = Excessive daytime sleepiness, was assessed by the Epworth sleepiness score 0–9 = 1, normal; 10–24 = 2, abnormal. ^3^ Sex: 0 = female, 1 = male. ^4^ Anxiety: HADS anxiety score 0–10 = 1, normal; 11–21 = 2, anxiety. ^5^ Depression: HADS depression score 0–10 = 1, normal; 11–21 = 2, depression. ^6^ Smoking 0 = no, 1 = yes. ^7^ Drinking 0 = no, 1 = yes, sometimes and often. ^8^ Exercise to sweating: 0 = no, 1 = yes. ^9^ Glycemic control: 0 = HbA_1_C ≤ 7.0, 1 = HbA_1_C > 7.0. ^10^ Lipid control: 0 = good, 1 = poor. ^11^ BP control: 0 = normal; 1 = >130/80 mmHg. ^12^ Diabetes control: 0 = normal, 1 = abnormal of glycemia, blood pressure, and dyslipidemia. ^13^ 4 complications: 0 = no, 1 = yes. ^14^ Complications: Number of complications, 0–4. Significant results were highlighted in bold.

**Table 5 ijerph-18-03025-t005:** Logistic regression on night time sleep quality and excessive daytime sleepiness.

	OR ^1^	95% CI		*p*
Poor night time sleep quality ^2^				
Age	0.99	0.956	1.025	0.589
Sex	0.29	0.147	0.584	0.000
Exercise to sweating	0.48	0.244	0.941	0.033
Ophthalmological problems	3.17	1.316	7.633	0.010
Vascular disease	1.35	0.681	2.681	0.389
	Nagelkerke R^2^ = 0.213			
Excessive daytime sleepiness ^3^				
Age	1.03	0.992	1.071	0.127
Sex	1.27	0.559	2.859	0.573
Sleep quality	4.35	1.765	10.732	0.001
Nephropathy	3.78	1.031	13.848	0.045
	Nagelkerke R^2^ = 0.164			

Note. ^1^ Wald test for Logistic regression, OR = odds ratio. ^2^ PSQI global score ≤5 = 1= good sleepers; >5 = 2 = poor sleepers. ^3^ Epworth sleepiness score <10 = 1 = normal; ≥10 = 2 = abnormal. Reference group: sex = female; exercise to sweating = no exercise, ophthalmological problems and vascular disease = no disease; sleep quality = good sleep; nephropathy = no symptom.

## Data Availability

The data presented in this study are available on request from the corresponding author. The data are not publicly available due to privacy.

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
