# Peer review of "Sleep Quality and Associated Factors in Adults with Type 2 Diabetes: A Retrospective Cohort Study"

_ijerph, 2021, doi:10.3390/ijerph18063025_

Round 1

Reviewer 1 Report

This work is a retrospective study in people with type 2 diabetes, in order to assess the sleep quality.

Interestingly, certain topics are related to diabetes control, for example zeitgeber (perception of light) which is an adapter for circadian rhythms that authors hypothesize that are worsened in the ophthalmological complications of diabetes.

I suggest in the abstract change the phrase “in type 2 diabetics” for “in persons with type 2 diabetes”.

Another minor topics are listed below:

Page 2, it appears a sign (>) in red colour. Please fix it.

Point 2.4.2, the ADA Guidelines revision (for the criteria of good metabolic control) is dated 2017. I suggest add the bibliographic reference and review for newest edition of the document. 

Table 1. The sign 75+ is confused, I think will be better >75 y. The same about diabetes duration 31+. Note that in the section of anxiety and depression is written as I suggest (>11 points, not 11+).

Author Response

Thank you for reviewers’ valued comments. We have revised the manuscript accordingly. Reviewers’ valued comments make this manuscript better. Please find our responses to reviewers point-by-point in the following. We have used the tracking function and the changed places we have marked them yellow. 

Manuscript ID: ijerph-1105264 - Major Revision

Title: Sleep Quality and Associated Factors in Adults with Type 2 Diabetes: A Retrospective Longitudinal Study.

To Reviewer 1:

Comments and Suggestions for Authors

This work is a retrospective study in people with type 2 diabetes, in order to assess the sleep quality.

Interestingly, certain topics are related to diabetes control, for example zeitgeber (perception of light) which is an adapter for circadian rhythms that authors hypothesize that are worsened in the ophthalmological complications of diabetes.

  1. I suggest in the abstract change the phrase “in type 2 diabetics” for “in persons with type 2 diabetes”.

Response: Thank you. We have revised it.

  1. Page 2, it appears a sign (>) in red colour. Please fix it.

Response: Thank you and have fixed it.

  1. Point 2.4.2, the ADA Guidelines revision (for the criteria of good metabolic control) is dated 2017. I suggest add the bibliographic reference and review for newest edition of the document.

Response: Thank you. We have updated to newest edition and added reference.

  1. Table 1. The sign 75+ is confused, I think will be better >75 y. The same about diabetes duration 31+. Note that in the section of anxiety and depression is written as I suggest (>11 points, not 11+).

Response: Thank you. We have revised accordingly.

Reviewer 2 Report

This paper was written with care and interest for sleep. I appreciate this topic and I believe that this paper can be made a bit more easier to read and interpret after these changes.

-Title : why was 'longitudinal' chosen as the data was collected cross-sectional? I would suggest deleting this word and replacing this with cross-sectional study

-methods : You have controlled for age in the analaysis. Was there a reason why you did not divide the entire group in age catagories <65y and +65y? As sleep will be different once people start ageing older. What is your motivation to not divide this group in to sub groups? Please take this up in your discussion

-methods : without mental disorders, such as....(it would be nice to the reader to complement this with examples and references)

-data collection : For logical reasons, I believe you should start with reporting demographics first (as you will discuss this first in results as well).

-Methods : The Cronbach's alpha 'sentence' is repeated twice. Please discard one sentence.

-Methods : it is not clear whether data was assessed at more than one point in time. Now it looks as if data were collected cross sectional (at one point). If so, please change accordingly in the title and in your methods section.

-Methods : it would be more logical to move 2.4.6 before 2.4.1

-data analysis : Why was logistic regression used? due to distributional reasons or other? I would suggest to motivate your choice for the reader in your paper.

Table 1 : the subcategories are now not so clearly indicated. I would suggest placing the words in bold (age, sex, ...) and have the text aligned to the left.

-results page 5 (bottom) : <7 hours of sleep (It woul be clear to the reader to mention whether naps were included in this sleep duration or not). And to take this up in your discussion as well.

-results, page 6 (bottom) : the greater the number of diabetic complications, the worse....

-Table 5 : it is not clear what Wald and EXP(B) stands for. I would suggest to take this up in the legend under the table

-discussion, thrice should be three times per week

-discussion, title 4.3 : performed intensive exercise : describe what is understood under intensive exercise (as well as in data collection under methods)

Author Response

Thank you for reviewers’ valued comments. We have revised the manuscript accordingly. Reviewers’ valued comments make this manuscript better. Please find our responses to reviewers point-by-point in the following. We have used the tracking function and the changed places we have marked them yellow. 

Manuscript ID: ijerph-1105264 - Major Revision

Title: Sleep Quality and Associated Factors in Adults with Type 2 Diabetes: A Retrospective Longitudinal Study.

Reviewer 2.

This paper was written with care and interest for sleep. I appreciate this topic and I believe that this paper can be made a bit more easier to read and interpret after these changes.

  1. Title: why was 'longitudinal' chosen as the data was collected cross-sectional? I would suggest deleting this word and replacing this with cross-sectional study

Response: Sleep data were surveyed once at recruitment in this study. However, status of diabetic control and complications were assessed by retrospectively reviewing patients’ medical records over 1 year prior to study enrollment. Therefore, we chose “A retrospective longitudinal study” for the title. We have clarified this in 2.3. Data collection session as well as in 2.4.3 and 2.4.4. Thank you for valued comments to make this paper clear.

  1. methods : You have controlled for age in the analaysis. Was there a reason why you did not divide the entire group in age catagories <65y and +65y? As sleep will be different once people start ageing older. What is your motivation to not divide this group in to sub groups? Please take this up in your discussion

Response: Thank you for valued comments. We have done the analysis with age, age divided into two groups (<65 and >=65 ) or age divided into 4 groups. Due to no significant difference within these grouping, we chose the simplest one (original age) to present. Since age is a very important factor on sleep, we still put age in the final model to control for. We have added related discussion in Discussion 4.3.

  1. methods : without mental disorders, such as....(it would be nice to the reader to complement this with examples and references)

Response: Thank you. We have reworded the inclusion and exclusion criteria in 2.1. “study design and participants” session.

  1. data collection : For logical reasons, I believe you should start with reporting demographics first (as you will discuss this first in results as well).

Response: Thank you. We have moved the sequence.

  1. Methods : The Cronbach's alpha 'sentence' is repeated twice. Please discard one sentence.

Response: Thank you. We have discarded the duplicated.

  1. Methods : it is not clear whether data was assessed at more than one point in time. Now it looks as if data were collected cross sectional (at one point). If so, please change accordingly in the title and in your methods section.

Response: Sleep data were surveyed once at recruitment in this study. However, status of diabetic control and complications were assessed by retrospectively reviewing patients’ medical records over 1 year prior to study enrollment. Therefore, we chose “A retrospective longitudinal study” for the title. We have clarified this in 2.3. Data collection session as well as in 2.4.3 and 2.4.4. Thank you for valued comments to make this paper clear.

  1. Methods : it would be more logical to move 2.4.6 before 2.4.1

Response: Thank you. We have removed the sequence.

  1. data analysis : Why was logistic regression used? due to distributional reasons or other? I would suggest to motivate your choice for the reader in your paper.

Response: Thank you for valued comments. Due to the sleep quality score (Global PSQI score) and the excessive daytime sleepiness score (ESS score) were not normally distributed in this study, logistic regression was selected for analysis in this study. We have added this explanation and data of the Kolmogorov-Smirnov test for normality for both PSQI and ESS score in the method 2.5 and result 3.3 sessions.  

  1. Table 1 : the subcategories are now not so clearly indicated. I would suggest placing the words in bold (age, sex, ...) and have the text aligned to the left.

Response: Thank you. We have replaced the text aligned to the left in Table 1 to 5.

  1. results page 5 (bottom) : <7 hours of sleep (It would be clear to the reader to mention whether naps were included in this sleep duration or not). And to take this up in your discussion as well.

Response: Thank you. Naps is not included in this sleep duration. We have added it in the result 3.2 session.

  1. results, page 6 (bottom) : the greater the number of diabetic complications, the worse....

Response: Thank you. Have added “the” accordingly.

  1. Table 5 : it is not clear what Wald and EXP(B) stands for. I would suggest to take this up in the legend under the table

Response: Thank you. We have explained that Wald test was for logistic regression and replaced EXP(B) with odds ratio (OR). Reference groups for each variable were indicated.

  1. discussion, thrice should be three times per week

Response: Thank you. Have revised.

  1. discussion, title 4.3 : performed intensive exercise : describe what is understood under intensive exercise (as well as in data collection under methods)

Response: Thank you. We have added the interpretation and implication for the result of intensive exercise (exercise to sweating) on sleep and diabetic control in the 4.3 discussion session. Definitions of lifestyle behaviors and sleep environment were also clarified in methods 2.4.5.  

Reviewer 3 Report

Review of “Sleep Quality and associated factors in Adults with Type 2 Diabetes: A Retrospective Longitudinal Study” (ijerph-1105264)

This study investigated the sleep quality of patients with T2DM and also investigated the factors which associated with sleep quality. This study revealed that lifestyle factors and diabetic complications are associated with sleep quality. This study is potentially interesting, however, several problems to be solved.

  1. Study design. This reviewer thinks that this study is not a retrospective longitudinal study, but a cross-sectional study.
  2. Pleases add the data of age, BMI, and duration, as continuous variables in Table 1.
  3. Furthermore, please add the data of ESS score and PSQI sore, as continuous variables.
  4. Previous studies revealed that “have to get up to use bathroom”, which was one of the major causes of disturbed nighttime sleep perceived, was associated with morality and diabetic complications in patients with T2DM. Furthermore, previous studies revealed that sleep quality was associated with QOL in patients with T2DM. The references below might be supporting this study.

Liu HY, et al. Int Urol Nephrol. 2016 Aug;48(8):1209-1214.

Furukawa S, et al. Urology. 2016 Jul;93:147-51.

Hashimoto Y, et al. BMC Endocr Disord. 2020 Jun 30;20(1):98.

Author Response

Thank you for reviewers’ valued comments. We have revised the manuscript accordingly. Reviewers’ valued comments make this manuscript better. Please find our responses to reviewers point-by-point in the following. We have used the tracking function and the changed places we have marked them yellow. 

Manuscript ID: ijerph-1105264 - Major Revision

Title: Sleep Quality and Associated Factors in Adults with Type 2 Diabetes: A Retrospective Longitudinal Study.

Reviewer 3.

Review of “Sleep Quality and associated factors in Adults with Type 2 Diabetes: A Retrospective Longitudinal Study” (ijerph-1105264)

This study investigated the sleep quality of patients with T2DM and also investigated the factors which associated with sleep quality. This study revealed that lifestyle factors and diabetic complications are associated with sleep quality. This study is potentially interesting, however, several problems to be solved.

  1. Study design. This reviewer thinks that this study is not a retrospective longitudinal study, but a cross-sectional study.

Response: Sleep data were surveyed once at recruitment in this study. However, status of diabetic control and complications were assessed by retrospectively reviewing patients’ medical records over 1 year prior to study enrollment. Therefore, we chose “A retrospective longitudinal study” for the title. We have clarified this in 2.3. Data collection session as well as in 2.4.3 and 2.4.4. Thank you for valued comments to make this paper clear.

  1. Pleases add the data of age, BMI, and duration, as continuous variables in Table 1.

Response: Thank you. We have added these data in Table 1.

  1. Furthermore, please add the data of ESS score and PSQI sore, as continuous variables.

Response: Thank you. We have added these data in Table 3.

  1. Previous studies revealed that “have to get up to use bathroom”, which was one of the major causes of disturbed nighttime sleep perceived, was associated with morality and diabetic complications in patients with T2DM. Furthermore, previous studies revealed that sleep quality was associated with QOL in patients with T2DM. The references below might be supporting this study.

Liu HY, et al. Int Urol Nephrol. 2016 Aug;48(8):1209-1214. Nocturia

Furukawa S, et al. Urology. 2016 Jul;93:147-51. Nocturia

Hashimoto Y, et al. BMC Endocr Disord. 2020 Jun 30;20(1):98.

Response: Thank you for valued comments and suggestion. We have revised the 4.1 discussion session regarding the association of nocturia and poor sleep quality with suggested citations.

Reviewer 4 Report

Manuscript entitled “Sleep Quality and Associated Factors in Adults with Type 2 Diabetes - A Retrospective Longitudinal Study” reports on factors associated with disturbed sleep in DM patients including general presentation of sleep quality in this group.

The abstract should be rewritten, particularly part presenting results, as it is unclear and confusing.

Exclusion criteria should be stated in separate sentence; present form makes it har to follow and understand.

Original articles on MMSE and HADS should be cited.

In describing each questionnaire, the reference to the original study should be provide.

Please clarity to which group individuals with 5 points in the PSQI were assigned.

On what bases the cutoff for ESS was chosen? As original interpretation is as follows:

0-5 Lower Normal Daytime Sleepiness

6-10 Higher Normal Daytime Sleepiness

11-12 Mild Excessive Daytime Sleepiness

13-15 Moderate Excessive Daytime Sleepiness

16-24 Severe Excessive Daytime Sleepiness.

Furthermore, the scale informs about ex essive daytime sleepiness, which is not only restricted to sleep apnea. It is also typical for narcolepsy and can be present fe in insomnia.

In describing each questionnaire, the reference to the original study should be provide.

Please clarity to which group individuals with 5 points in the PSQI were assigned. Typically results equal or higher than 6 are considered as poor sleep.

When referring to the American Diabetes Association Guidelines please add a citation of the paper.

Complication of DM should also be supported by a literature (citation).

It would add additional information and perspective dividing the drop into controlled and uncontrolled DM, compare the groups and perform multiple regression to assess the risk factors (if any sleep parameters are associated with poor glycemia control). Additionally, the study participants could be divided into poor and good sleepers and compered followed by multiple regression to assess the risk factors of poor sleep.

Data distribution should be assessed and if it does not have normal distribution the parameter should be presented by media and IQR instead of mean and SD.

Cutoff points for BMI are not standard please either used standard division, or explain the why such cutoff points were used.

I would advise to add more literature describing association between T2DM and sleep disorders explaining additional mechanism responsible to this relationship (see: 10.1016/j.diabres.2010.07.011).

Author Response

Thank you for reviewers’ valued comments. We have revised the manuscript accordingly. Reviewers’ valued comments make this manuscript better. Please find our responses to reviewers point-by-point in the following. We have used the tracking function and the changed places we have marked them yellow. 

Manuscript ID: ijerph-1105264 - Major Revision

Title: Sleep Quality and Associated Factors in Adults with Type 2 Diabetes: A Retrospective Longitudinal Study.

Reviewer 4.

Manuscript entitled “Sleep Quality and Associated Factors in Adults with Type 2 Diabetes - A Retrospective Longitudinal Study” reports on factors associated with disturbed sleep in DM patients including general presentation of sleep quality in this group.

  1. The abstract should be rewritten, particularly part presenting results, as it is unclear and confusing.

Response: Thank you. Have revised the abstract especial the result part.

  1. Exclusion criteria should be stated in separate sentence; present form makes it har to follow and understand.

Response: Thank you. We have reworded the inclusion and exclusion criteria in 2.1. “study design and participants” session.

  1. Original articles on MMSE and HADS should be cited.

Response: Thank you. Have provided citations in 2.1 (MMSE) and 2.4.6 (HADS).

  1. In describing each questionnaire, the reference to the original study should be provide.

Response: Thank you. Have provided the citations in 2.4 Measure session.

  1. Please clarity to which group individuals with 5 points in the PSQI were assigned.

Typically results equal or higher than 6 are considered as poor sleep.

Response: Sorry for confusing. Yes, PSQI score <=5 was assigned as “good” and >5 (>=6) as “poor.” We have clarified and corrected it in 2.4.2.

  1. On what bases the cutoff for ESS was chosen? As original interpretation is as follows:

0-5 Lower Normal Daytime Sleepiness

6-10 Higher Normal Daytime Sleepiness

11-12 Mild Excessive Daytime Sleepiness

13-15 Moderate Excessive Daytime Sleepiness

16-24 Severe Excessive Daytime Sleepiness.

Response: Thank you for this valued and important comment. We know the cutoff for ESS was suggested as the above. However, due to the performance of ESS in our population (Chen et al, 2002 as below), we chose 10 as the cutoff. Therefore, we used this criterion for data analysis. We have revised the related data in Method 2.4.2, Table 3, Table 4, and Table 5 to make it clear.  

Chen, N.H., Johns, M. W., Li, H. Y., Chu, C. C., Liang, S. C., Shu, Y. H., Chuang, M. L., Wang, P. C.et al., Validation of a Chinese version of the Epworth sleepiness scale. Qual Life Res, 2002. 11(8): p. 817-21 doi: 10.1023/a:1020818417949.

  1. Furthermore, the scale informs about ex essive daytime sleepiness, which is not only restricted to sleep apnea. It is also typical for narcolepsy and can be present in insomnia.

Response: Thank you for valued comments. Yes, ESS >15 is a risk for narcolepsy. There were 4 patients with ESS score > 15 in this study. We have added this information in 4.2 discussion.

  1. When referring to the American Diabetes Association Guidelines please add a citation of the paper.

Response: Thank you. We have updated to newest edition and added reference.

  1. Complication of DM should also be supported by a literature (citation).

Response: Thank you. We have added reference in Method 2.4.4 session.

  1. It would add additional information and perspective dividing the drop into controlled and uncontrolled DM, compare the groups and perform multiple regression to assess the risk factors (if any sleep parameters are associated with poor glycemia control). Additionally, the study participants could be divided into poor and good sleepers and compered followed by multiple regression to assess the risk factors of poor sleep.

Response: Thank you for valued comments and suggestions. We have replaced the Pearson correlation results with univariate logistic regression with odds ratio in Table 4 and revised the 3.3 result session. The multiple logistic regression is followed based on the result of univariate analysis. 

  1. Data distribution should be assessed and if it does not have normal distribution the parameter should be presented by median and IQR instead of mean and SD.

Response: Thank you. We have also added median and IQR data for PSQI sleep score and EDS score in Table 3.

  1. Cutoff points for BMI are not standard please either used standard division, or explain the why such cutoff points were used.

Response: Thank you for valued comments. We used Asian standard and recommendations from Taiwan Ministry of Health and Welfare for the BMI cutoff points. References have been provided and clarified in 2.4.1 Measures.

  1. I would advise to add more literature describing association between T2DM and sleep disorders explaining additional mechanism responsible to this relationship (see: 10.1016/j.diabres.2010.07.011).

Response: Thank you for valued comments. Yes, we have cited this reference and describing the association between T2DM and sleep disorder in the Introduction and Discussion session.

Round 2

Reviewer 3 Report

The authors revised well. However, I cannot agree to study design.  In this study, they did not observe the outcome measures more than twice repeatedly.

Author Response

Dear Editor,

Thank you for reviewers’ valued comments. We have revised the manuscript accordingly. Reviewers’ valued comments make this manuscript better. Please find our responses to reviewers point-by-point in the following. We have used the tracking function and the changed places we have marked them yellow.

Regarding reviwer’s #3 report about study design:

Data of diabetic control status and complications were measured and retrieved from medical records every three months retrospectively all over 1 year at the study enrollment for three to four time points. The sleep quality and other data were measured at enrollment. The sequence of study data was that diabetic control and complications happened first, then sleep quality last. After short discussing, Dr Lee and I carefully revise the manuscript in title and method session about the study design as a retrospective cohort study. Thank you for valued comments to improve this paper.

Reviewer 4 Report

Author addressed the comments well and improved manuscript.

Please expand abbreviation used in the tables in their description.

In the discussion regarding possible mechanism involved in relationship between sleep and DM2 consider doi: 10.3389/fphys.2020.01035.

Author Response

Reviewer 4

Author addressed the comments well and improved manuscript.

Please expand abbreviation used in the tables in their description.

Response: Thank you. We have checked all the tables and corrected the abbreviation in Table 2.

In the discussion regarding possible mechanism involved in relationship between sleep and DM2 consider doi: 10.3389/fphys.2020.01035.

Response: Thank you. We have added the possible mechanism in sleep and T2DM in the Discussion session 4.2 and cited the above reference.